# Cranial Spinal Spreading of Canine Brain Gliomas after Hypofractionated Volumetric-Modulated Arc Radiotherapy and Concomitant Temozolomide Chemotherapy: A Four-Case Report

**DOI:** 10.3390/vetsci9100541

**Published:** 2022-09-30

**Authors:** Gaetano Urso, Alexandra Gabriela Boncu, Nancy Carrara, Dragos-Teodor Zaman, Luca Malfassi, Silvia Marcarini, Lucia Minoli, Simone Pavesi, Massimo Sala, Eugenio Scanziani, Mario Dolera

**Affiliations:** 1Azienda Socio Sanitaria Territoriale di Lodi, 26900 Lodi, Italy; 2La Cittadina Fondazione Studi e Ricerche Veterinarie, 26014 Romanengo, Italy; 3Dipartimento di Scienze Veterinarie, Università degli Studi di Torino, 10095 Grugliasco, Italy; 4Dipartimento di Medicina Veterinaria e Scienze Animali, Università degli Studi di Milano, 20122 Milano, Italy; 5Mouse and Animal Pathology Lab (MAPLab), Università degli Studi di Milano–La Statale–Fondazione UniMi, 20122 Milano, Italy

**Keywords:** dog gliomas, volumetric-modulated radiotherapy, temozolomide, spinal-spreading

## Abstract

**Simple Summary:**

Gliomas represent the second-most-common primary brain tumors in dogs. Surgery and radiotherapy are established treatment approaches with similar median survival time, whereas conventional chemotherapy is burdened by severe adverse effects. Spinal and leptomeningeal spread of gliomas have been described following radiotherapy treatment alone and the prognosis is generally inauspicious. In this work, a more aggressive radiotherapy protocol with concomitant radiosensitizing chemotherapy is represented but spinal-spread of the disease is reported, despite the fact that it was originally located exclusively in the brain area. This work suggests that the entire central nervous system should be investigated in diagnostic examinations and during the follow-up of canine gliomas. A radiotherapy dose-escalation trial and different chemotherapy dosages on a larger number of animals could estimate the optimal protocol to avoid spinal-spread while limiting radio-chemo toxicities. In the same context, prophylactic treatment to the spinal cord with a lower dose could also be the key to preventing recurrence across the cranial–spinal pathway and to improving quality of life.

**Abstract:**

Gliomas are the second-most-common primary brain tumors in dogs. Surgery and radiotherapy are established treatment approaches with similar median survival time, whereas conventional chemotherapy is burdened by severe adverse effects. Spinal and leptomeningeal spread of gliomas have been described following radiotherapy treatment alone. The purpose of this study was to evaluate the outcome for four dogs with primary high-grade gliomas in the forebrain without evidence, at diagnosis, of neoplastic invasion along the spinal cord, that were treated with concomitant chemotherapy (temozolomide) and hypofractionated volumetric-modulated arc radiotherapy (VMAT-RT). Temozolomide was selected for its radiosensitive properties, and radiotherapy dose protocols of 37 Gy in 7 fractions or 42 Gy in 10 fractions were used. After an initial complete or partial response, tumors recurred across the cranial–spinal pathway. Post-mortem macroscopic examinations confirmed swollen spinal cord and hyperemic meningeal sleeve, with nodular lesions on the meningeal surface. Microscopically, infiltration of the spinal cord and meninges by neoplastic cells (with features of oligodendrogliomas) were observed. This work seems to suggest that the entire central nervous system should be investigated in diagnostic examinations of canine gliomas. Dose-escalation trials and/or spinal cord prophylaxis treatment could also be evaluated to prevent tumor progression.

## 1. Introduction

The prevalence of primary intracranial neoplasms in dogs is estimated to be 2.3% in the whole canine population, with meningiomas being the most common, especially in dolichocephalic breeds [1]. Gliomas represent the second-most-common primary intracranial neoplasm, accounting for 36.6% of primary brain tumors and are most commonly diagnosed in brachycephalic breeds [1]. Gliomas are a heterogeneous group of neoplasms, including astrocytomas, oligodendrogliomas, oligoastrocytomas, glioblastomas, and gliomatosis cerebri, all arising from the supporting cells of the brain parenchyma [1]. Increasing accessibility to advanced diagnostic imaging tools, such as magnetic resonance imaging (MRI) and computed tomography (CT), has improved the ability to identify these types of tumors [2]. Studies in the last decade have shown a median age at diagnosis of eight years, with a strong relationship between age and body weight [1,3]. Although gliomas are diagnosed more frequently in larger breeds and older dogs, astrocytomas can be found in young dogs, similar to what is observed in human pediatric patients [4]. Breeds such as boxers, Boston terriers, French and English bulldogs, English toy spaniels and bullmastiffs appear to be predisposed to developing brain tumors [1,5,6]; a proposed hypothesis for this higher incidence is altered intracranial pressure due to skull anatomy, and decreased ability of the brain to handle pressure changes [7,8,9]. Recently, Bannasch et al. [6], using a different approach to analyze genome traits of various breeds, identified strong genome-wide associations for brachycephalic head type on cyclopropane fatty acid synthesis (Cfa); similarly, Truve et al. [10] identified three genes highly associated with glioma susceptibility (CAMKK2, P2RX7, and DENR) that share the same locus with genes used for selection in brachycephalic dog breeds.

Treatment approaches for intracranial tumors and, in particular, gliomas are surgical resection, chemotherapy, and radiation therapy (RT) or, at the dog owner’s option, a symptomatic approach to improve the pet’s quality of life [11,12]. In a recent comprehensive review [12] of brain-tumor treatment in dogs, the authors concluded that median survival time (MST) after either surgery or radiotherapy, as the leading treatments was similar (312 and 351 days, respectively). Chemotherapy does not appear to give survival advantages over symptomatic treatment (MST of 75 and 65 days, respectively), and is burdened by serious adverse effects, especially bone-marrow suppression (thrombocytopenia and leukopenia) [12,13]. Taking into consideration the often-inoperable location of gliomas, RT appears to be a viable option when choosing cytoreductive treatment [14,15,16].

In human medicine, the so called “drop metastasis” (diffuse spinal and leptomeningeal spreading of gliomas) has been described in some cases [17,18,19,20,21,22,23,24,25,26,27], and it occurs mainly in cases of very aggressive high-grade gliomas, such as multiform glioblastoma or other anaplastic astrocytomas. In veterinary medicine, canine drop metastasis has been reported [28] but, to the best of our knowledge, only a single case-study reports systematic spinal-spread after exclusive radiation therapy of a primary intracranial mass classified histologically as a leptomeningeal oligodendroglioma [29]. Another recent paper [30] describes the MRI findings of cerebrospinal fluid (CSF) drop metastasis due to canine gliomas according to the human classification system [31].

The purpose of this article is to describe four cases of dogs with primary brain tumours (glioma) with no initial spinal involvement recurring in the spinal cord (via the cranio-spinal route) after an initial complete or partial response, despite an aggressive combined treatment of the primary tumor consisting of hypofractionated volumetric-modulated arc radiotherapy (VMAT) and concomitant chemotherapy (temozolomide, TMZ).

This work seems to suggest that a prophylactic treatment to the spinal cord with a lower dose could prevent tumor progression and/or improve quality of life.

## 2. Materials and Methods

Four dogs with a presumed brain glioma were considered for this report. For all dogs included in the study, informed consent for the entire treatment and the possibility of using the data for research and published works was obtained from the owners before any clinical evaluation. In all cases, the basic neurological examinations and diagnoses were performed by the same clinician (M.D.), based on the modified Glasgow Coma Scale [32] and subsequent MRI of the whole central nervous system (CNS). After the diagnosis, the dogs were irradiated by hypofractionated volumetric-modulated radiotherapy (VMAT) with concomitant and adjuvant temozolomide.

The MRI examinations and the CT simulation were conducted as described elsewhere [33]. The presumptive diagnosis of glioma was made on the relationship between MRI criteria and tumor grade according to the 2007 World Health Organization (WHO) Classification of Tumours of the Central Nervous System [34].

For the delineation of the target volume, the MRI was fused with the planning CT using a manual tracing method. Image fusion between the MRI and CT scans was performed with a semi-automatized protocol on a graphic table (Wacom, London, UK) using dedicated software (Monaco; CMS Elekta, Stockholm, Sweden).

For well-demarcated tumors, gross tumor volume (GTV) was defined by contouring the area of signal and structure abnormality. For infiltrating tumors, the GTV encompassed all areas of signal abnormality on the same sequences. Planning target volume (PTV) was obtained through an isotropic expansion of 2 mm from the GTV. The organs at risk (OARs) were the eyes, the optic pathways, the basal ganglia, the cerebrum, the hypothalamus, the pituitary gland, the brainstem, the cerebellum, the spinal cord, the inner ear and the trachea. The OAR dose constraints were derived from those reported by the American Association of Physicists in Medicine Task Group 101 [35].

Corticosteroids and anticonvulsant were administered to all the dogs as described elsewhere [32].

Radiotherapy treatment was administered using a 6 MV photon beam through an Elekta Synergy S linear accelerator equipped with a micro multileaf beam collimator (Elekta Beam Modulator (EBM)) and an XVI cone beam CT system (CBCT). VMAT treatments were planned using a Monte Carlo statistical algorithm and the (Elekta, CMS Stockholm, Sweden) Monaco treatment-planning system (version 3.0 or above). Setup verification was performed through the XVI system before each fraction. The RT protocol was as follows: 37 Gy in 7 fractions in two dogs and 42 Gy in 10 fractions in the others two patients. In detail, a total dose of 42 Gy was chosen for tumors of greater relative volume and/or when their localization led to greater involvement of organs at risk.

The plan was evaluated by way of standard dose volume histograms (DVH). Specifically, the degree of PTV coverage considered acceptable for the V95% and V107% levels (the PTV volume receiving less than 95% and more than 107% of the dose prescription) was 5% and 1%, respectively. Radiotherapy treatment was combined with TMZ, administered orally at the dose of 65 mg/m^2^ six h prior to each RT fraction and then for five days monthly for six cycles.

Neurological and clinical examinations were performed daily during the irradiation treatment period and then weekly for the first month. Serial MRI examinations were performed 2 months after irradiation and then at 4, 6, 9, 12, 18 and 24 months. Moreover, if indicated by the clinical condition, additional MRI examinations were acquired. For comparison purposes, all MRI scans were performed with the same scanner, and with the parameters used at the time of diagnosis. The volumetric modifications were analyzed on transverse T2-W images using the previously described manual tracing method by the same clinician that assessed the radiotherapy CT simulation tumor volume, and variations were classified using the Response Evaluation Criteria In Solid Tumours (RECIST) system [36].

The Veterinary Radiation Therapy Oncology Group (VRTOG) criteria [37] were used to evaluate and grade radiation toxicities while the Veterinary Cooperative Oncology Group—Common Terminology Criteria for Adverse Events (VCOG-CTCAE) [38] was used for analyzing the TMZ-related adverse effects.

Due to progression of the tumor and deterioration of the animals’ health, all four dogs were euthanatized, and pathological examination was performed, with macroscopic and histopathological evaluation of the cervical region of the spinal cord and the meninges. For histopathology assessment, post-mortem samples were fixed in formalin, paraffin-embedded and 4 µm thick sections stained with hematoxylin and eosin. To confirm the morphological diagnosis, in one case, immunohistochemistry for Iba1 (marker of macrophages and microglia), glial fibrillary acidic protein (GFAP, marker of astrocytes), and Olig2 (marker of oligodendrocytes) was performed.

## 3. Results

### 3.1. Case Descriptions

The signalment and the main clinical data of the dogs included in this report are reported in Table 1

All the four dogs presented with generalized seizures. Neurological examination was normal in two dogs (cases 1 and 2) and abnormal with proprioceptive deficits in the other two (cases 3 and 4). Physical, physiological, and laboratory workups were normal in all dogs.

In all cases, an intra-axial solitary mass was detected in the forebrain with no evidence of any neoplastic tissue along the spinal cord or other organs of interest. The lesions were circumscribed, oval shaped, with sharp margins. Although deeply localized in the brain, the tumors reached the sub-arachnoid space. For all the dogs, during follow-up examinations, progressive paresthesia, paraparesis, and cervical pain were observed.

In detail, cases 1 and 2 presented a delimited nodular lesion in the frontal lobe, with features of grade III presumed glioma, T2-W hyperintensity, and T1-W hypointensity with slight contrast enhancement. One month post the RT treatment, a near-complete response with low-contrast enhancement was recorded in both cases. Case 1 presented at the third control (after 6 months) a thin blade-like hyperintensity lesion located in the spinal cord, with enlargement of both the third and fourth ventricles. Nine months after RT (fourth control) the above-described MR features had visibly worsened in addition to a noticeable hyperintensity on the cervical part of the spinal cord, meningeal thickening with contrast enhancement, and a significant swelling of the central canal with severe ependymitis. In case 2 similar pathological changes were observed four months after the end of the RT course.

Cases 3 and 4 presented a large temporo-paraventricular grade IV presumed glioma with necrosis, hemorrhage, and cystic regions with contrast enhancement in case 3 (Figure 1a). Two months after the radiotherapy course, both dogs had a near-complete response with no sign of recurrence and no contrast enhancement (Figure 1b).

In both cases, the neurological status improved to normal and seizures gradually subsided in terms of both severity and frequency. Consequently, the corticosteroid treatment was gradually tapered. Six months after RT, case 3 presented a residual gliotic scar at the primary tumor site (Figure 1c). Nine months after RT, the sagittal T2-W MRI showed the progression of the disease along the entire spinal cord (Figure 1d) with a hyperintensity signal evidenced in the cervical and thoracic tract of the spinal cord, which was swollen with a concomitant enlargement of the fourth ventricle and the central canal, and hyperintensity of the meninges. Cerebrospinal fluid (CSF) examination ruled out inflammatory processes and infections, and the presence of neoplastic cells was compatible with a cranio-spinal diffusion of the tumor.

### 3.2. Post-Mortem Examination

The post-mortem macroscopic and histological assessment showed similar findings in all cases. The macroscopic evaluation confirmed spinal cord swelling, with hyperemic meningeal sleeve, and nodular lesions on the meningeal surface. Microscopically, leptomeninges and the underlying spinal cord were expanded and infiltrated by a densely cellular neoplasia compatible with an oligodendroglioma (Figure 2a).

The same neoplastic population invaded the ependymal canal and infiltrated the surrounding nervous tissue (Figure 2b). Tumor cells were round to polygonal with lightly stained cytoplasm and central round hyperchromatic nucleus. (Figure 2c). Within the tumor, multifocal, lobulated microvascular proliferations (glomeruloid bodies) arranged in long lines and clusters were visible (Figure 2d). In one case (case 3), immunohistochemistry was performed to define tumor histotype. Tumor cells were negative for glial fibrillary acidic protein (GFAP, marker of astrocytes) and ionized calcium-binding adapter molecule (Iba1, marker of microglial cells), in the presence of inner positive controls (Figure 2e,f). Further, the neoplastic population had diffuse nuclear expression of Olig2, a marker of oligodendrocytes (Figure 2g,h).

## 4. Discussion

In this paper, we describe four cases of complete or partial response of primary oligodendrogliomas treated with RT and concomitant temozolomide and the subsequent spreading of the tumor along the spinal cord. The primary tumor was originally located in the forebrain in all four cases without evidence of neoplastic tissue along the spinal cord. As a differential diagnosis, another form of glioma, gliomatosis cerebri (GC), was considered. Cerebral gliomatosis is a rare type of neoplasm that has been described to infiltrate the spinal cord in dogs [39,40], but, differently from our cases, MRI examination showed that cerebral gliomatosis tended to be widespread, with relative conservation of neuronal architecture [40,41,42,43], predominantly astrocytic, and without neovascularization [44,45]. In the dogs reported in the present paper, the primary lesions were well circumscribed, did not occupy three continuous lobes, were not present in the thalamus, and the area of the caudal fossa was not affected. In our cases, the presence of a single lesion on the frontal or temporal lobes with sharp margins and with a clear distinction between grey matter and white matter favored the diagnosis of glioma. Furthermore, post-mortem histological examination definitely excluded the differential diagnosis of cerebral gliomatosis.

Following the treatment and the initial partial or complete response of the tumor, the follow up MRI evidenced swelling of the cervical spinal cord with hyperintense T2-W signal, and meningeal thickening with contrast enhancement in all the dogs. In one dog, the whole spinal cord was involved, while the other three exhibited cervical spreading and neoplastic infiltration was demonstrated by post mortem histological examinations.

Similar to the cases in this study, primary high-grade gliomas in humans tend to occupy primarily in the cerebrum with very few exceptions (such as the lateral ventricles and cerebellum), whereas spinal-spread is more often diffuse or intramedullary, and very rarely occupies extramedullary space [26]. Dissemination of primary intracranial tumors mostly occurs in the subarachnoid space through the CSF or through the central canal [46,47,48], but dissemination through the Virchow–Robin perivascular space has also been suggested [49]. Another hypothesized mechanism of tumor-cell propagation in gliomas, particularly in glioblastoma associated with the highest degree of vessel formation, is vascular invasion [50]. Vascular endothelial growth factors have been shown to be overexpressed in glioblastoma and to promote intense angiogenesis [51]. Finally, another possible mechanism suggested for tumor-spreading in irradiated glioblastoma is through the glymphatic system, organized with functional lymphatic vessels lining the dural sinuses, transporting both fluid and immune cells from the CSF, connected to the deep cervical lymph nodes [52]. The glymphatic system is facilitated by aquaporin 4, a water-channel protein found in preclinical models on the astrocytes’ end-feet along blood vessels in the brain and spinal cord, which allows the CSF to flow into the perivascular space and interstitial fluid [53].

In human oncology, most glioma-related metastases occur after surgical removal of the primary tumor and adjuvant therapies. It has been suggested that tumor-cell manipulation [21,26,54], radiation therapy, and perhaps concomitant treatment with subtherapeutic doses of temozolomide may contribute to tumor recurrence and/or spread [55,56,57,58].

In agreement with this theory, in the present work we hypothesized that, even in veterinary cases, following a combined chemo-radiotherapy treatment the loss of oligodendroglioma cell arrangement could contribute to dissemination due to a subcytotoxic chemotherapy dosage administered or due to the radiotherapy fractionation dose scheme. There is also evidence that the concentration of TMZ achieved within the CSF is cytotoxic only to cancer stem cells negative for methylation status, whereas methylation-positive stem cells and differentiated cancer cells are resistant [58]. Studies performed on humans suggest that methylation can modify the sensitivity of glioblastoma cells to TMZ [59]. In particular, promoter methylation of the O-6-methylguanine-DNA methyltransferase (MGMT) gene is considered to be a predictive biomarker for TMZ-mediated benefits in glioblastoma [60], with an improvement in the progression-free survival (PFS) observed in 39.7% of patients with MGMT promoter-methylated tumors vs. 6.9% without MGMT promoter methylation [61].

By analyzing the spread of oligodendrogliomas along CSF pathways at multiple sites within the brain and spinal cord in three dogs, Koch et al. [62] suggested that oligodendroglioma should be a differential diagnosis for multifocal lesions, especially if the lesions were near or contiguous with ventricular spaces. They also reported that MRI alone may be insufficient to make a diagnosis of intracranial oligodendrogliomas even though some imaging findings, particularly T2-weighted hyperintensity along with T1-weighted hypointensity, were strongly suggestive of these tumors.

In a more recent veterinary medicine case report on a female English bulldog diagnosed with grade III multifocal oligodendrogliomas [29], the authors described histological and immunohistochemical features similar to those we presented in the present article (rounded neoplastic cells with hyperchromatic nucleus expressing positivity to Olig2). The intracranial, lateral ventricle mass shrank after stereotactic ablative radiotherapy (SABR) delivered using 6 MV photon beam with an intensity-modulated radiotherapy treatment and three fractions of 6 Gy on alternating weekdays; the dog developed a central vestibular syndrome after 5 weeks; a new MRI and CT scan showed intramedullary lesions at C1–C5 level and involvement of the ventral side of the leptomeninges at the C7 level. The new presumptive diagnosis was CSF drop metastasis in the cervical central canal and leptomeninges. A single fraction of 600 cGy using matched parallel-opposed fields was prescribed to the entire central nervous system. The dog was hospitalized until returning to preanesthesia status the next day. The dog expired several hours after discharge, 147 days after the first SABR. The presumed cause of death was respiratory failure due to severe cervical myelopathy.

In a recent article [30], ten dogs with histologically confirmed glioma foci were analyzed and divided into two groups: the first group (group “A”; seven dogs) presented initially with solitary lesions that subsequently metastasized. Cases in group “B” (three dogs) had multifocal lesions (primary and metastatic) at first presentation. The dogs in group “A” received different first-option treatments; in particular, one dog received a fractionated radiotherapy (FRT) with total 50 Gy dose in 20 fractions while a second dog was irradiated with stereotactic radiotherapy (SRT). All cases, additionally, received symptomatic therapy (glucocorticoids and anticonvulsant drugs), and two received enteral chemotherapy. Progressive disease was treated in three dogs (SRT for local recurrence in two cases; surgery then whole-central-nervous-system (CNS) radiotherapy for a total of 30 Gy in 10 fractions for local recurrence and CSF drop metastasis in one case; one dog also received intrathecal infusion of QUAD-doxorubicin for intraspinal drop metastasis). The SRT doses ranged between 15 and 24 Gy in 1–3 fractions and the dogs died in 96–681 days. In the group “B”, a symptomatic or a uPA-targeted oncolytic Newcastle virus (three intravenous injections) treatment was erogated; the dogs were euthanized between the day after the MRI diagnosis and 135 days after the treatment. In all three dogs, complete necropsy revealed intracranial and spinal glioma. However, the goal of the reported retrospective multi-institutional study [30] was to describe the MRI findings of canine glioma CSF drop metastasis and categorize them according to the human classification system [31]. Based on the two canine case-reports [30,63], the authors of this study [30] hypothesized that the metastatic lesions would have different imaging characteristics compared to the primary tumor. To our knowledge, based on a regular review of the literature, there was only one clinical case by Nakamoto et al. where a dog was treated with a combined radiotherapy and chemotherapy protocol [63]. This was applied to a 2.5 years old male French pit bull euthanized after 356 days; the authors suggested that the possibility of leptomeningeal dissemination and hydrocephalus should be considered even after RT and chemotherapy for anaplastic oligodendroglioma. In the present work, a combined approach of VMAT radiotherapy and chemotherapy with a radiosensitizing agent was chosen, but with a different fractionation scheme, an almost homogenous target dose coverage, and a higher equivalent dose, with a hypothesized result of improved local tumor control. In particular the dose scheme was selected for maximizing the dose to the target while maintaining the dose to the OAR below the limits described elsewhere [35]. The total dose delivered in the study by Nakamoto et al. [63] was of 49 Gy given in 7 fractions but with 95% of PTV volume covered by a dose of 28 Gy. In the present paper the total dose ranged from 37 to 42 Gy delivered in 7 and 10 fractions, respectively, but the treatment was administered with a volumetric-modulated intensity technique characterized by an homogeneous dose with the 95% PTV volume covered by a minimum dose of 35.2 Gy and 39.9 Gy (95% of the prescribed dose), respectively. For comparing the radiotherapy dose prescription with the work from Nakamoto et al., the alpha/beta ratio for the tumor was derived from the human ones [64]. In detail, using a mean value for the alpha/beta reported for gliomas, a 7.7 value was chosen; nominally, the 37 Gy/7 fractions and 42 Gy/10 fractions radiotherapy dose scheme used in the present work resulted in biological equivalent doses (BEDs) of 62.4 Gy and 64.9 Gy, respectively, while the 49 Gy/7 fractions used by Nakamoto resulted in a BED of 93.6 Gy. Nevertheless, considering PTV coverage at 95% dose, an effective BED of 42.6 Gy was obtained by Nakamoto et al., while in the present work BEDs resulted in 58.2 Gy and 60.6 Gy for prescribed doses of 35.2 Gy and 39.9 Gy. The dosage and chemotherapeutic agents were also different from those of the study of Nakamoto et al., which was based on a lomustine dosage of 30–60 mg/m^2^ administered every 3 weeks, followed by maintenance chemotherapy with the same agent at a dosage of 60 mg/m^2^ every 4 weeks, whereas in the present work TMZ was chosen for its radiosensitizing properties, although no net benefit of the use of this agent was demonstrated with regard to MST, but only on progression-free survival time [33].

In addition, also in the work of Bentley et al. [30], one dog received radiotherapy treatment with a nominal BED of 66.2 Gy (50 Gy in 20 fractions) similar to those prescribed in the present work while other dogs received stereotactic radiotherapy with doses in the range of 15–24 Gy with 1–3 fractions but none of them received combination radiotherapy and chemotherapy as a first approach.

The cases reported in the present paper seem to suggest that, both at baseline and during the follow up, the MRI should not be limited to the brain, but should be extended to the whole CNS, to evidence, in a timely manner, possible involvement of the spinal cord.

Moreover, the absence of severe side effects showed the safety of the protocol and could be related to the high conformity obtained using the VMAT technique, which permitted good target coverage while sparing the OAR, suggesting dose-escalation trials with a possible improvement in tumor-control probability (TCP).

Despite a combined approach of radiotherapy and chemotherapy, a cranio-spinal diffusion was observed in the present work, suggesting the possibility of more aggressive chemotherapy dosage and radiotherapy fractionation and/or the opportunity of a lower radiotherapy dose treatment as prophylaxis on the spinal cord.

## 5. Conclusions

The combined treatment with VMAT and TMZ reported in this work was well tolerated. However, after partial or complete response, primary oligodendroglioma confined to the brain could recur in the spinal cord.

Although TMZ can be used as a radiosensitizer without serious side effects, a dose-escalation study with volumetric-modulated arc radiotherapy on a larger number of animals could estimate the optimal dose to avoid spinal-spread by limiting radiotoxicity.

In the same context, prophylactic treatment of the spinal cord at low doses could also be evaluated to prevent tumor progression and improve quality of life.

In addition, in all diagnostic examinations related to canine gliomas, the entire CNS should be investigated.

## Figures and Tables

**Figure 1 vetsci-09-00541-f001:**
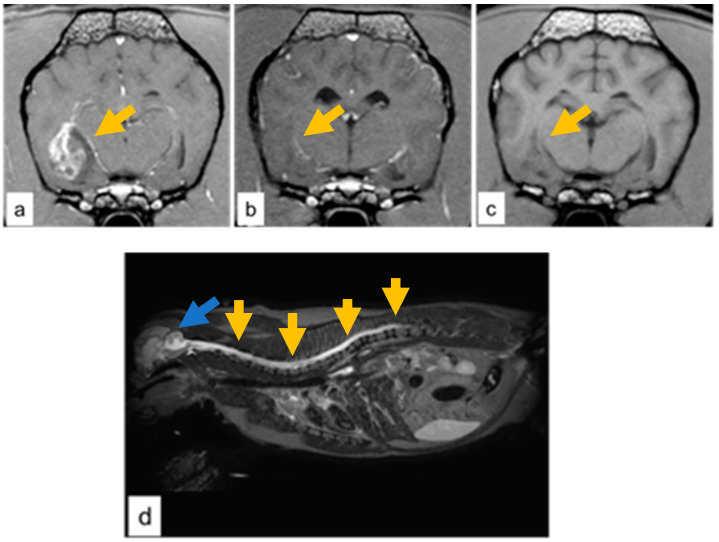
Case 3, grade IV glioma (**a**) T1-W image with contrast enhancement showing a well-demarcated tumor at the temporal-paraventricular region (dark-yellow arrow). (**b**) Two months post RT near-complete regression was observed (dark-yellow arrow). (**c**) Six months post RT, hypointense lesion consistent with a gliotic scar at the temporal-paraventricular region (dark-yellow arrow). (**d**) Case 4, grade IV glioma; cervical and thoracic involvement of the spinal cord on T2-W image by the hyperintensity signal (dark-yellow arrow). and with concomitant enlargement of the fourth ventricle (blue arrow).

**Figure 2 vetsci-09-00541-f002:**
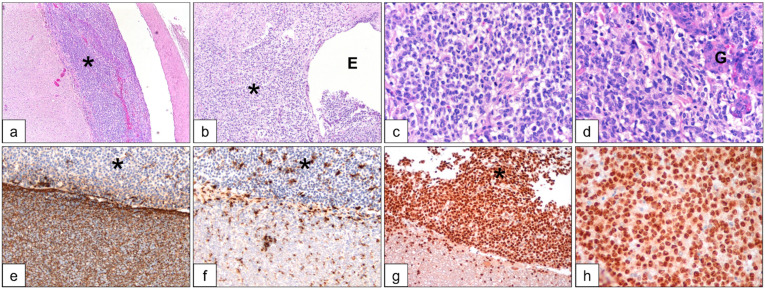
Case 3, histological and immunohistochemistry findings of the dissected neoplastic tissue in the spinal cord. H&E stained slides (**a**–**d**) show neoplastic cells (*) at the meningeal area and at the ependymal canal (**E**), multifocally forming glomeruloid bodies (**G**). By immunohistochemistry, neoplastic cells (*) were negative for GFAP (**e**) and Iba1 (**f**), while positive for Olig2 (**g**,**h**).

**Table 1 vetsci-09-00541-t001:** Signalment and main clinical data.

Dog Number	Signalment	Diagnosis	Signs	Neurological Examination	Best Response	First Event	Time to First Event
1	Boxer, female, 8 years old	Grade III glioma	Generalized seizures	Normal	a	Spinal cord lesion	6 months
2	French bulldog, male, 8 years old	Grade III glioma	Generalized seizures	Normal	a	Spinal cord lesion	4 months
3	French bulldog, male, 7 years old	Grade IV glioma	Generalized seizures	Proprioceptive deficits	b	Gliotic scar at primary tumor site	6 months
4	Labrador, female, 8 years old	Grade IV glioma	Generalized seizures	Proprioceptive deficits	b	Spinal cord lesion	9 months

a = near complete response at 1 month; b = near complete response at 2 months.

## Data Availability

The data presented in this study are available on request from the corresponding author. The data are not publicly available due to the fact that they have not yet been published and are, for the most part, comprised of images.

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
