# Peer review of "Cranial Spinal Spreading of Canine Brain Gliomas after Hypofractionated Volumetric-Modulated Arc Radiotherapy and Concomitant Temozolomide Chemotherapy: A Four-Case Report"

_vetsci, 2022, doi:10.3390/vetsci9100541_

Round 1

Reviewer 1 Report

I consider this to be a very interesting piece of research, specifically for the veterinary area and for the pharmaceutical industry, since it requires the development of new therapies aimed at destroying tumors, such as nanotechnology, without affecting benign cells.

I consider that the work is very well cited and everything is explained very well. However, the work can be published after certain corrections.

1. Is there any ethical consent or permission for the studies performed on the 4 dogs?

2. In the case of figures 1 A, B, C, D, it would be better to represent what each image indicates by means of arrows for a better understanding of the reader since not all readers will be veterinarians.

3. The same for figure 2, mark with arrows what each image indicates for a better understanding of the reader

4. Would you as researchers recommend treatment with temozolomide?

5. I think the conclusion could be expanded a little more

6. I respectfully suggest adding a couple of updated references from either 2020 or 2021.

Author Response

  1. Is there any ethical consent or permission for the studies performed on the 4 dogs?

Yes: upon editor request, between the first submission and the reviewers observations, we have sent a blank version of it (so you had not it but, now, we have submitted it). Moreover, we have added this sentence in the M&M section: “For all dogs included in the study, informed consent was obtained from the owners on the entire treatment and on the possibility of using the data for research and published works before any clinical evaluation.” and we have modified the Informed Consent Statement in this way: “Informed consent was obtained from all subjects involved in the study.”

  1. In the case of figures 1 A, B, C, D, it would be better to represent what each image indicates by means of arrows for a better understanding of the reader since not all readers will be veterinarians.

We agree and we have modified the figure 1 accordingly.

  1. The same for figure 2, mark with arrows what each image indicates for a better understanding of the reader

The images are already marked with asterisks. If you retain that asterisks are not visible, we may add an arrow.

  1. Would you as researchers recommend treatment with temozolomide?

Yes, of course, since it acts as a radiosensitizer. Moreover, even if  the mean survival time doesn’t change significally as reported in another paper from our group ( “Frameless stereotactic radiotherapy alone and combined with temozolomide for presumed canine gliomas” - Vet Comp Oncol . 2018 Mar;16(1):90-101. doi: 10.1111/vco.12316), with temozolomide the free survival time is longer than without. We have modified/added a sentence both into the “Discussion” and the “Conclusions” sections (so we have expanded a little more them) as follows:

(Discussions):  “The dosage and chemotherapeutic agents were also different from the study of Nakamoto et al., which was based on a lomustine dosage of 30-60 mg/m2 administered every 3 weeks, followed by maintenance chemotherapy with the same agent at a dosage of 60 mg/m2 every 4 weeks, whereas in the present work TMZ was chosen for its radiosensitizing properties, although no net benefit of the use of this agent was demonstrated with regard to MST, but only on free survival time.”

(Conclusions):  “Although TMZ can be used as a radiosensitizer without serious side effects, a dose esca-lation study with volumetric modulated arc radiotherapy on a larger number of animals could estimate the optimal dose to avoid spinal spread by limiting radiotoxicity.”

  1. I think the conclusion could be expanded a little more

We have rephrased the Conclusion in this way:

“The combined treatment with VMAT and TMZ reported in this work was well tolerated. However, after partial or complete response, primary oligodendroglioma confined to the brain could recur in the spinal cord. 

Although TMZ can be used as a radiosensitizer without serious side effects, a dose escalation study with volumetric modulated arc radiotherapy on a larger number of animals could estimate the optimal dose to avoid spinal spread by limiting radiotoxicity.

In the same context, prophylactic treatment of the spinal cord at low doses could also be evaluated to prevent tumor progression and improve quality of life.

In addition, in all diagnostic examinations related to canine gliomas, the entire CNS should be investigated.”

We could add something about comparison with recent literature, but having only four cases we preferred to write nonspecific sentences. Do you think we should add something about the comparison?

  1. I respectfully suggest adding a couple of updated references from either 2020 or 2021.

We  have checked literature in pubmed with keywords but we have found just four papers.

 We have added these references (and reported them into the paper):

[16] Debreuque, M.; De Fornel, P.; David, I.; Delisle, F.; Ducerveau, M.N.; Devauchelle, P.; Thibaud, J.L. Definitive-intent uniform megavoltage fractioned radiotherapy protocol for presumed canine intracranial gliomas: retrospective analysis of survival and prognostic factors in 38 cases (2013–2019). BMC Vet. Res. 2020, 412. https://doi.org/10.1186/s12917-020-02614-x

[28] Rohrer Bley C, Staudinger C, Bley T, Marconato L, Sabattini S, Beckmann K. Canine presumed glial brain tumours treated with radiotherapy: Is there an inferior outcome in tumours contacting the subventricular zone? Vet Comp Oncol. 2022 Mar;20(1):29-37. doi: 10.1111/vco.12703. Epub 2021 May 10. PMID: 33900018.

[39] Giron, C.; Paquette, D.; Culang, D.; Doré, M.; Masseau, I. Diffuse meningeal oligodendrogliomatosis characterized by spinal intra-parenchymal nodules on magnetic resonance imaging in a dog. Can. Vet. J. 2020, 61(12):1312-1318. PMID: 33299250; PMCID: PMC7659878.

[40] Rissi, D.R.; Donovan, T.A.; Porter, B.F.; Frank, C.; Miller, A.D. Canine Gliomatosis Cerebri: Morphologic and im-munohistochemical characterization is supportive of glial histogenesis. Vet. Pathol. 2021, 58(2):293-304. doi: 10.1177/0300985820980704. Epub 2020 Dec 28. PMID: 33357125.

We have added also this reference that we have cited has part of the answer to your point 4:

[32] Dolera, M.; Malfassi, L.; Bianchi, C.; Carrara, N.; Finesso, S.; Marcarini, S.; et al. Frameless stereotactic radiotherapy alone and com-bined with temozolomide for presumed canine gliomas. Vet. Comp. Oncol. 2018, 90-101. https://onlinelibrary.wiley.com/doi/abs/10.1111/vco.12316

Reviewer 2 Report

The introduction states that chemotherapy is not of benefit in these types of cases, so the reader wonders why it was included (line 64-65 versus line 78-82).

Please also make it clear in lines 78-82 that all 4 cases were primary brain tumors with subsequent spinal involvement.

It is very important for this manuscript and for the reader, please stress the entire CNS was evaluated (line 92-93).

Is it appropriate to change "epileptic seizures" to "generalized seizures" throughout the manuscript?

It is very important for the reader to know the clinical signs, not just MRI findings, in the follow up of each patient.  Were clinical signs evident?  Case 1?  Case 2 - was there only cervical pain at all times?  Were there clinical signs in Case 3?  Case 4 had cervical pain and paraparesis. 

What medical therapy was provided to each case?  Line 210-211 stated that corticosteroid therapy was tapered  When was it started?  Were all cases treated with corticosteroids?  Other medications?

Line 292-205:  Is loss of oligodenroglioma cell arrangement known to occur with TMZ?  The remainder of the paragraph describes concentrations in the CSF and methylation of cells.

The paragraph starting on line 322 - what dog is described in lines 322 - 326?  One previously reported?  If yes, please reference.  If not, please explain its inclusion in the discussion.

Lines 354-373 - how does the approach of this manuscript thought to have affected the cases, if it was different than previously reported treatments?  The paragraph describes the differences from the previous literature but does not explain how the difference was chosen and how it could change the outcome in the cases reported here.

The conclusion in lines 395-398 seems beyond the scope of this case series.

Author Response

The introduction states that chemotherapy is not of benefit in these types of cases, so the reader wonders why it was included (line 64-65 versus line 78-82).

In lines 64-65 we made an excursus on the treatment modalities reported in literature while the lines 78-82 report the approach proposed in this paper. In literature, there hasn’t been reported a combined approach and the chemotherapy exclusive treatment doesn’t seem to give a benefit over symptomatic treatment. Since the prognosis, for this type of disease, is inauspicious, a more aggressive treatment with locoregional radiotherapy and chemotherapy (with temozolamide that acts also as a radiosensitizer) was attempted in the present work in an attempt to prevent spinal spread of the disease. In a previous version of the paper, this was “clearly”: do you retain that we have to insert some sentences? In this case, where do you suggest? It could be included in both the introduction and the conclusion, but this could make the speech heavier. Please: feel free to give us your suggestions.

Please also make it clear in lines 78-82 that all 4 cases were primary brain tumors with subsequent spinal involvement.

We have modified the sentence in this way: “The purpose of this article is to describe four cases of dogs with primary brain tumours (glioma) with no initial spinal involvement recurred in the spinal cord (via the craniospinal route) after an initial complete or partial response, despite an aggressive combined treatment of the primary tumor consisting of hypofractionated volumetric arc modulated radiotherapy (VMAT) and concomitant chemotherapy (temozolomide, TMZ).”

It is very important for this manuscript and for the reader, please stress the entire CNS was evaluated (line 92-93).

We added "on the whole central nervous system" in this way: " In all cases, the basic neurological examinations and diagnoses were performed by the same clinician (M.D. ), based on the modified Glasgow Coma Scale [32] and subsequent MRI of the whole central nervous system (CNS)," because this is the M&M section, so it is important for the reader to know that the whole CNS was studied, but it is in the results section (lines 180-181) that it was reported (and, earlier, also in the introduction) that initially the spinal cord was not involved. Do you agree?

Is it appropriate to change "epileptic seizures" to "generalized seizures" throughout the manuscript?

We agree: we haven’t observed focal seizure but generalized. We changed throughout the manuscript

It is very important for the reader to know the clinical signs, not just MRI findings, in the follow up of each patient.  Were clinical signs evident?  Case 1?  Case 2 - was there only cervical pain at all times?  Were there clinical signs in Case 3?  Case 4 had cervical pain and paraparesis.

In all dogs we noted cervical pain, progressive paresthesia that gradually turns into tetraparesis. We reported paraperesis separately for case 4 because the tumor was spreaded all along the spinal cord, imprinting compression on the thoracic region of the spinal cord while, for case two, we wanted to say that the study was conducted upon observation and request of the owner but was not significant. We changed the sentence structure and wrote a new one involving all dogs regarding clinical signs and eliminated the part about them in the case discussions, where only MRI findings are reported.

The sentence has been added just before the detailed description of each case as reported below:

“For all the dogs, during follow up examinations, progressive paraesthesia, paraperesis and cervical pain were observed.”

What medical therapy was provided to each case?  Line 210-211 stated that corticosteroid therapy was tapered  When was it started?  Were all cases treated with corticosteroids?  Other medications?

Because all dogs received the same ancillary treatment, we added these sentences to the M&M section:

“Corticosteroids and anticonvulsant were administered to all the dogs as described elsewhere [32].”

I had originally written the following sentence (in green), but I had the problem of single singularity with respect to a previous work of mine, so I changed it (and also two other sentences in the M&M sections): “Corticosteroids (Medrol vet, Pfizer Italia, Ascoli Piceno, Italy or Desashok, Fort Dodge, Milan, Italy) were administered to all the dogs, starting just before the first radiotherapy session, at a dosage ranging from 0.5 mg/kg to maximum 2 mg/kg twice daily.

All the cases received also one or more of the following anticonvulsant: Phenobarbital (Gardenale; Aventis Pharma, Milan, Italy), at variable doses, starting from 2 mg/kg to maximum 4 mg/kg twice daily; topiramate (Topamax; Janssen-Cilag, Milan, Italy) from 5 mg/kg to 10 mg/kg twice daily; levetiracetam (Keppra; UCB Pharma, Milan, Italy) 20 mg/kg twice daily or three times a day.”

It’s clear that I could change the sentences but the technical data are that… (just as the VCOG title cited in the paper)

Line 292-205:  Is loss of oligodenroglioma cell arrangement known to occur with TMZ?  The remainder of the paragraph describes concentrations in the CSF and methylation of cells.

In our opinion, TMZ acts on cells but not on mutual "messages." We believe that the loss of dispositions of oligodendroglioma cells may be related to radiotherapy and not to TMZ, but it is only a hypothesis and we do not feel comfortable to go into the discussion in the text even because TMZ acts also as radiosensitizer so the whole effect is not well known in this case and we are not sure on the single component of the treatment that can modify the cell arrangement. Indeed, if you feel that we have made excessive assumptions, we can revise the sentences

The paragraph starting on line 322 - what dog is described in lines 322 - 326?  One previously reported?  If yes, please reference.  If not, please explain its inclusion in the discussion.

It has not to be a new paragraph since the dog described is the English bulldog from the ref. 29. It has been included because the authors described histological and immunohistochemical features similar to those we presented in the present article and we have reported the treatment outcome for comparison (there are very few papers on this topic) but if you retain that we should summarize and remove some parts, we’ll take into account your considerations

Lines 354-373 - how does the approach of this manuscript thought to have affected the cases, if it was different than previously reported treatments?  The paragraph describes the differences from the previous literature but does not explain how the difference was chosen and how it could change the outcome in the cases reported here.

First of all, we want to apologize for some typo: the 37 Gy/7 fractions dose scheme results in biological equivalent doses (BEDs) of 64.2 Gy (and not 70.6 Gy) and considering PTV coverage at 95% dose, an effective BED of 60.6 Gy (and not 60.1 Gy) is obtaind for prescribed dose of 39.9 Gy. We have modified the text.

The TMZ was chosen for its radiosensitizer properties. We have added two sentences answering to a comment from the other reviewer that I want to report to you below in green

  1. Would you as researchers recommend treatment with temozolomide?

Yes, of course, since it acts as a radiosensitizer. Moreover, even if  the mean survival time doesn’t change significally as reported in another paper from our group ( “Frameless stereotactic radiotherapy alone and combined with temozolomide for presumed canine gliomas” - Vet Comp Oncol . 2018 Mar;16(1):90-101. doi: 10.1111/vco.12316), with temozolomide the free survival time is longer than without. We have modified/added a sentence both into the “Discussion” and the “Conclusions” sections (so we have expanded a little more them) as follows:

(Discussions):  “The dosage and chemotherapeutic agents were also different from the study of Nakamoto et al., which was based on a lomustine dosage of 30-60 mg/m2 administered every 3 weeks, followed by maintenance chemotherapy with the same agent at a dosage of 60 mg/m2 every 4 weeks, whereas in the present work TMZ was chosen for its radiosensitizing properties, although no net benefit of the use of this agent was demonstrated with regard to MST, but only on free survival time.”

(Conclusions):  “Although TMZ can be used as a radiosensitizer without serious side effects, a dose esca-lation study with volumetric modulated arc radiotherapy on a larger number of animals could estimate the optimal dose to avoid spinal spread by limiting radiotoxicity.”

Moreover, our protocol it’s different from the other since the dose schemes lead to higher BED and VMAT treatments is characterized by an higher level of homogeneity and, in this way, by an higher mean dose to the target. These futures, could give a better tumor control but we have observed, anyway, spinal diffusion so the tumor control is only local. We have added this sentence into the “Discussions” section:

“In the present work, a combined approach of VMAT radiotherapy and chemotherapy with a radiosensitizing agent was chosen, but with a different fractionation scheme, an almost homogeneus target dose coverage and a higher equivalent dose, with a hy-pothesized result of improved local tumor control. In particular the dose scheme has been selected for maximizing the dose to the target while maintaining the dose to the OAR below the limits described elsewhere [35].”

The conclusion in lines 395-398 seems beyond the scope of this case series.

The sentence explains what we are, currently, doing at our facility. In a sense, it could be placed in a section: "Future Developments" and thus goes beyond the scope of this paper, as you wrote. We felt, however, it was correct to include it as a natural conclusion to the report. If you feel it should be deleted, we will delete it.

Please, take into account that we have modified the “Conclusion” section answering to a comment from the other reviewer as follow:

“The combined treatment with VMAT and TMZ reported in this work was well tolerated. However, after partial or complete response, primary oligodendroglioma confined to the brain could recur in the spinal cord. 

Although TMZ can be used as a radiosensitizer without serious side effects, a dose esca-lation study with volumetric modulated arc radiotherapy on a larger number of animals could estimate the optimal dose to avoid spinal spread by limiting radiotoxicity.

In the same context, prophylactic treatment of the spinal cord at low doses could also be evaluated to prevent tumor progression and improve quality of life.

In addition, in all diagnostic examinations related to canine gliomas, the entire CNS should be investigated.”